# Vitamin D in Central Nervous System: Implications for Neurological Disorders

**DOI:** 10.3390/ijms25147809

**Published:** 2024-07-17

**Authors:** Bayan Sailike, Zhadyra Onzhanova, Burkitkan Akbay, Tursonjan Tokay, Ferdinand Molnár

**Affiliations:** Department of Biology, School of Sciences and Humanities, Nazarbayev University, Kabanbay Batyr 53, Astana 010000, Kazakhstan; bayan.sailike@nu.edu.kz (B.S.); zhadyra.onzhanova@nu.edu.kz (Z.O.); burkitkan.akbay@nu.edu.kz (B.A.); tursonjan.tokay@nu.edu.kz (T.T.)

**Keywords:** calcitriol, vitamin D, brain, neurological disorders

## Abstract

Vitamin D, obtained from diet or synthesized internally as cholecalciferol and ergocalciferol, influences bodily functions through its most active metabolite and the vitamin D receptor. Recent research has uncovered multiple roles for vitamin D in the central nervous system, impacting neural development and maturation, regulating the dopaminergic system, and controlling the synthesis of neural growth factors. This review thoroughly examines these connections and investigates the consequences of vitamin D deficiency in neurological disorders, particularly neurodegenerative diseases. The potential benefits of vitamin D supplementation in alleviating symptoms of these diseases are evaluated alongside a discussion of the controversial findings from previous intervention studies. The importance of interpreting these results cautiously is emphasised. Furthermore, the article proposes that additional randomised and well-designed trials are essential for gaining a deeper understanding of the potential therapeutic advantages of vitamin D supplementation for neurological disorders. Ultimately, this review highlights the critical role of vitamin D in neurological well-being and highlights the need for further research to enhance our understanding of its function in the brain.

## 1. Introduction

Vitamin D (VD) is a fat-soluble steroid hormone mainly existing in two isoforms: cholecalciferol (VD_3_) and ergocalciferol (VD_2_). VD_3_ can be synthesised in the skin when exposed to sunlight (Figure 1A) or absorbed from fortified products, such as eggs, oily fish, dairy products, and meat (Figure 1B) [1]. During the first step of VD_3_ biosynthesis, the penetration of ultraviolet B (UVB) radiation with a wavelength of 290–315 nm into epidermis stimulates the photo-conversion of 7-dehydrocholesterol into pre-VD_3_ (precalciferol) and subsequently calciol (Figure 1A), which is then metabolised by hydroxylation reactions catalysed by cytochrome P450 monooxygenases (CYP) CYP2R1 and CYP27A1 in the liver to form 25-hydroxyvitamin D_3_ (calcidiol), as seen in Figure 1C. Calcidiol is further hydroxylated by calcidiol-1α -hydroxylase (CYP27B1) to 1α,25-dihydroxyvitamin D_3_ (calcitriol), the most active form of VD_3_ in the body, as seen in Figure 1D,E [2,3]. Although the kidneys are the primary site for the synthesis of circulating active calcitriol, recent research indicates that several other cell types, including prostate, placenta, brain, lung, immune cells, and myocytes, that also express 1α-hydroxylase, enable the local conversion of calcidiol to its active form [4,5].

The biological actions of calcitriol are mediated through the vitamin D receptor (VDR), which is a transcription factor regulating gene expression, as seen in Figure 2. The VDR is a member of the nuclear receptor (NR) family that include receptors for retinoic acid, thyroid hormone, steroid hormones, and adrenal steroids [6]. Once calcitriol binds to the VDR, it forms a heterodimer with another NR, the retinoic acid X receptor (RXR), and then translocates to the nucleus, resulting in the VD response elements (VDREs) recognition, binding, and subsequent target gene activation, as seen in Figure 2A. The calcitriol/VDR/RXR/VDREs complex regulates the transcription of many genes, including the activation of *calbindin* (*CALB*), *cathelicidin* (*CAMP*), *transforming growth factor β* (*TGFB*), and *nerve growth factor* (*NGF*) as well as the repression of *parathyroid hormone* (*PTH*) and *CYP27B1*, as seen in Figure 2 [7].

It has been known for almost a century that the central role of VD_3_ is the mineral homeostasis and regulation of bone growth [8]. In addition to its classical roles in bone metabolism, growing evidence shows that VD_3_ exerts other essential physiological functions in the human body. For instance, it regulates the proliferation, differentiation, and apoptosis of normal and malignant cells [9]. Moreover, the expression of the *VDR* and *CYP27B1* in certain types of immune cells implies the potential roles of VD_3_ in the modulation of the immune function [10]. Furthermore, calcitriol, as a neurosteroid hormone, exerts various actions in the central nervous system (CNS): it affects brain development, maintains adult brain function, and protects the brain from aging [3,11]. It is also involved in the regulation of neurotrophins, neuroimmunity, and neurotransmission in the brain [12]. Moreover, numerous studies have found a link between VD_3_ deficiency and several neurological conditions such as autism, schizophrenia, multiple sclerosis, Parkinson’s disease, and Alzheimer’s diseases, leading to the hypothesis that VD_3_ may play a vital role in the pathogeneses of neurological diseases [13,14]. Therefore, this review aims to discuss the function of VD_3_ in the CNS and its role in the pathophysiology of different neurological disorders. The studies we referenced were conducted using calcitriol (unless stated otherwise), the active form of VD_3_, due to its direct biological actions, particularly in relation to cellular functions in the CNS. We focused on calcitriol rather than cholecalciferol or calcifediol, as these forms undergo further metabolic transformations before becoming biologically active. This distinction is crucial, as the effects of vitamin D on cellular interactions differ significantly depending on the form utilised with calcitriol being immediately active upon administration.

## 2. The Effect of VD_3_ on the Developing Brain

The enzymes which are linked to the biosynthesis and metabolism of calcitriol, as well as the VDR, are present in various neural cells such as oligodendrocytes, astrocytes, microglia, and neurons [15], suggesting that calcitriol is formed locally in the brain tissue, as seen in Figure 1E. In addition, VD_3_ and its metabolites can cross the blood–brain barrier (BBB) [16]. Therefore, the possible role of VD_3_ in CNS has gained prominent attention, and its effects have been extensively studied. One of these studies is the effect of VD_3_ on the brain development.

A number of in vivo and in vitro studies have demonstrated that VD_3_ has anti-proliferative, pro-differentiation and pro-apoptotic effects on the brain cell by regulating genes (Table 1) [17]. For example, when rats were deficient in VD_3_ during gestation, cell proliferation was enhanced and apoptosis was reduced in multiple regions of the brain, leading to abnormal development of the CNS [18]. The increased proliferation might be mediated by the up-regulation of *cyclin D1* (*CCND1*) gene expression and down-regulation of cyclin-dependent kinase inhibitor 1A *CDKN1A or p21*, while the reduction in apoptosis may be caused by decreased levels of *B-cell lymphoma 2 apoptosis regulator* (*BCL2*) and *BCL2 Antagonist/Killer 1* (*BAK1*) genes [18]. in vitro studies indicate that the addition of VD_3_ inhibits the proliferation of cultured hippocampal neurons while promoting neurite outgrowth [19]. This effect may be due to the up-regulation of *NGF* by VD_3_. Additionally, VD_3_ stimulates the differentiation of neural stem cells into neurons and oligodendrocytes [20]. Consistent with this, it was shown that oligodendrocyte precursor cells are unable to differentiate when VDR signalling is inhibited [21]. Therefore, it is likely that VD_3_ functions as a critical regulator for oligodendrocytes and neuron development. Possibly, the induction of neurotrophins may be the underlying mechanism for the effect of VD_3_ on neural stem cells and oligodendrocyte differentiation [20,21]. VD_3_ deficiency decreases *NGF*, *glial-derived neurotrophic factor* (*GDNF*), and the *nerve growth factor receptor* (*NGFR*) in neonatal rat brains [17]. In addition, the neural stem cells treated with VD_3_ up-regulate the expression of *brain-derived neurotrophic factor* (*BDNF*), *neurotrophin-3* (*NTF3*), *GDNF*, and *ciliary neurotrophic factor* (*CNTF*) [20].

VD_3_ also affects the development of the dopaminergic system [3]. A significant finding suggests that VD_3_ is vital for the early differentiation of dopaminergic neurons with important implications for schizophrenia research [22]. in vitro studies show that VD_3_ participates in the differentiation of DA neurons, influencing the expression of the key enzymes of the DA production pathway, such as *tyrosine hydroxylase* (*TH*) and *catechol-O-methyltransferase* (*COMT*) [23]. in vivo studies confirmed that VD_3_ deficiency modifies DA signalling by regulating proto-oncogene *RET* expression [24]. Other studies have shown that VD_3_ deficiency during development of the foetal brain exerts adverse effects on developing DA systems. The absence of VD_3_ during development results in a significant decrease in the expression of DA-specific factors such as *nuclear receptor related 1 protein* (*NURR1/NR4A2)*, *TH*, and *cyclin dependent kinase inhibitor 1C* (*CDKN1C*), which are the important differentiation markers of DA neurons [25,26]. A recent study using animal models showed that supplementation with calcitriol could rescue the expression of DA-specific factors such as *Lmx1a* and enhance the DA cell number in mesencephalic cells [27]. It is also able to reinstate the abnormalities in the dopaminergic system that have been improperly induced by prenatal maternal immune activation [27]. Moreover, a recent study by Pertile et al. has demonstrated that VD_3_ has the capacity to regulate the differentiation and function of DA. Their findings revealed that VD_3_ promotes neurite outgrowth and induces alterations in the distributionand expression of synaptic vesicle proteins that are involved in DA release within these neurites. This study utilised three distinct in vitro models, including the SH-SY5Y cell line, primary mesencephalic cell culture, and rat embryonic striatal-mesencephalic explant co-culture [22].
ijms-25-07809-t001_Table 1Table 1The effect of VD_3_/calcitriol on the function of brain cells (↑—up-regulated/increased, ↓—down-regulated/decreased).Model/StudiesVD_3_ DeficiencyVD_3_ SupplementationGene(s) InvolvedStudyAnimal/in vivo**↑ Proliferation in neonatal rat brain**
↓ *NGF, GDNF, NGFR*[17]Animal/in vivo**↑ Proliferation during gestation in the rat**
↑ *CCND1*/↓ *CDKN1A*[18]Animal/in vivo**↓ Apoptosis during gestation in the rat**
↓ *BAK1, BCL2*[18]Cellular/in vitro
**↑ Hippocampal neurite outgrowth**↑ *NGF*[19]Neural stem cells/in vitro
**↑ Differentiation to neurons and oligodendrocytes**↑ *BDNF, NTF3, GDNF, CNTF*[20]Cellular/in vitro
**↑ Differentiation of dopaminergic neurons in SH-SY5Y cells**↑ *TH, COMT*[23]Animal and cellular/in vivo and in vitro**↑ Dopaminergic neuronal survival in****VD_3_-deficient rat model and SH-SY5Y**
↓ *RET*[24]Animal/in vivo**↓ Dopaminergic neuronal differentiation and maintenance VD_3_-deficient rat mesencephalon**
↓ *TH, NR4A2, CDKN1C*[25,26]Animal/in vivo**↓ Neuronal plasticity in VD_3_-deficient rat brain**
↓ *GAP43*[28]Animal/in vivo
**↑ Hippocampal synaptic function in aging rats**↑ *SYNJ1, SYT2, CAMK2D*[29]Cellular/in vitro
**↑ Neuroprotection in hippocampal neurons**↓ *CACNA1C, CACNA1D*[30]Cellular/in vitro
**↓ Free radicals and ROS production in neurons and microglia**↓ *NOS2*, Activity: ↓ NFκB and ↑ GGT↑ GGT activity[31]Cellular/in vitro
**↑ Immune response in microglial cell line**↓ *IL6, IL1B, TNF, IFNG, CCL3*↑ *TGFB1, IL10, IL4, IFNA, IFNB*[32,33]Cellular/in vitro
**↑ Immune response in astrocyte**↓ *TNF, IL1, TLR4*[34]

There is evidence suggesting that VD_3_ plays a role in neuronal plasticity. In a study on prenatal VD_3_-deficient rat brains, researchers observed a dysregulation of synaptic plasticity-related genes, including *growth-associated protein 43 (GAP43)* [28]. This finding suggests that VD_3_ is crucial for the normal development and function of the CNS. Furthermore, supplementation with VD_3_ has been proposed to stabilise myelin structure, enhance synaptic vesicle recycling, and promote the activity of transcription factors involved in cognitive processes, which essentially contribute to preservation during aging. The enhanced expression of multiple genes essential for synaptic plasticity include for instance *synaptojanin 1* (*SYNJ1*), a phosphatase that partly regulates the clathrin-coated vesicles endocytosis or recycling, *synaptotagmin 2* (*SYT2*), which after calcium (Ca^2+^) binding aids in the docking of filled vesicles, and *calcium/calmodulin-dependent protein kinase IIδ* (*CAMK2D*), which is initiated on the postsynaptic neurons after the stimulation of glutamate or dopamine receptors. Subsequently, this leads to the activation of transcription factors *cAMP response element binding protein 3* (*CREB3*) and *NR4A2* promoting the expression of genes related to memory processes [29].

VD_3_ also contributes to the maintenance of intracellular Ca^2+^ homeostasis in neurons and glial cells [5]. VD_3_ down-regulates L-type voltage-gated calcium channel subunit α 1 C and D (CACNA1C, CACNA1D), leading to a decrease in excitotoxicity injury in hippocampal neurons triggered by a sudden increase of cytoplasmic Ca^2+^, which could cause an elevation in the reactive oxygen species (ROS) level and mitochondrial dysfunction, eventually causing neuronal cell death [30]. VD_3_ may reduce the formation of free radicals and production of ROS in neurons and microglia by decreasing the synthesis of *inducible nitric oxide synthase* (*NOS2*), reducing the activity of nuclear factor κ-light-chain-enhancer of activated B cells (NFκB), and increasing the activity of the γ-glutamyl transferase (GGT) [31].

Finally, VD_3_ plays a pivotal role in regulating the immune system within the brain by modulating gene expression. in vitro studies have demonstrated that treatment with VD_3_ leads to a decrease in the levels of pro-inflammatory cytokines, such as *interleukin 6* (*IL6*), *interleukin 1 beta* (*IL1B*), *tumour necrosis factor* (*TNF*), *interferon gamma* (*IFNG*), and *C-C motif chemokine ligand 3 CCL3*, while promoting the expression of anti-inflammatory cytokines, including *transforming growth factor beta 1* (*TGFB1*), *interleukin 10* (*IL10*), *interleukin 4* (*IL4*), *interferon alpha* (*IFNA*), and *interferon beta* (*IFNB*) in the microglia cell line [32,33]. Moreover, in vitro studies have revealed that VD_3_ significantly reduces the expression of pro-inflammatory cytokines such as *TNF* and *IL1* as well as *toll-like receptor 4* (*TLR4*) in activated astrocytes [35]. This suggests that VD_3_ plays a crucial role in mitigating astrocyte-mediated inflammation. In addition, a combination treatment of VD_3_ and progesterone has shown a decrease in *TLR4* expression in the CNS of mice with traumatic brain injury, highlighting the potential therapeutic implications of VD_3_ in reducing inflammation in the context of brain damage [34]. Taking together the above highlighted studies collectively emphasises the immunoregulatory effects of VD_3_ in different cell types within the brain, including microglia and astrocytes. The ability of VD_3_ to modulate inflammatory responses suggests its potential as a therapeutic agent for neuroinflammatory conditions and highlights the importance of further investigating its immunomodulatory role in maintaining brain health.

## 3. VD_3_ and Schizophrenia

Schizophrenia is a group of neuropsychiatric disorders characterised by positive symptoms including hallucinations and delusions, negative symptoms, such as depression, impaired motivation, affective flattening, and cognitive deficits [36]. The development of schizophrenia is complex, and it is driven by genetic risks interacting with multiple vulnerability factors [37]. In both animal and human studies, VD_3_ deficiency has been identified as a plausible risk factor for the development of schizophrenia [37]. Animal experiments have demonstrated that transient prenatal VD_3_ deficiency is associated with enduring changes in brain structure and function [38]. Similarly, in patients with schizophrenia, VD_3_ deficiency has been shown to alter DA function [27]. Yee et al. conducted a study examining VD_3_ levels in first-episode psychosis patients [39]. In this study, patients with schizophrenia exhibited low levels of VD_3_, which were also associated with negative symptoms. Although the Finnish birth cohort study found an increased risk of schizophrenia in male infants born without VD_3_ supplementation during the first year of life, conducting clinical trials to examine the effect of maternal VD_3_ supplementation on the incidence of schizophrenia in offspring has been challenging due to plausibility and ethical concerns [40]. Furthermore, clinical intervention studies were also conducted to assess if VD_3_ supplementation could improve the syndromes of patients with schizophrenia. One study showed that 50,000 IU of VD_3_ supplementation per week for a duration of 12 weeks alleviates the symptoms and improves metabolic profiles [41]. On the contrary, another clinical trial showed that VD_3_ supplementation with 300,000 IU per month via an intramuscular injection of VD_3_ twice within 3 months in patients maintaining an anti-psychotic treatment regime did not find any significant improvement in symptoms [42]. Fond et al. have shown that 12 months of VD_3_ supplementation in schizophrenic patients without hospitalisation is associated with lower depressive symptoms and lower rates of anxiety [43], whereas earlier studies have shown that 2000 IU per day of orally administered VD_3_ for 8 weeks did not help with any improvement in symptoms [44]. Recently, dysregulation of the cellular redox environment as well as inflammation in patients with schizophrenia has attracted attention regarding the underlying mechanism of VD3 action (Figure 3). Lower glutathione (GSH) levels were observed in vivo in the prefrontal cortex of schizophrenia patients [45,46]. Moreover, GSH insufficiency caused by redox dysregulation can result in N-methyl-D-aspartate receptor (NMDAR) hypo-function and reduced myelination [47,48]. VD_3_ is thought to regulate the gene expression and activation of *GGT*, thereby controlling the synthesis of GSH [49]. Furthermore, VD_3_ acts as a neuroprotective agent that can reduce damage to neurons and oligodendrocytes by down-regulating *iNOS*, which catalyses the synthesis of nitric oxide (NO) [50]. Consistent with this finding, elevated levels of NO and iNOS were detected in patients with schizophrenia in post-mortem studies [51,52,53]. There is also evidence indicating adverse immune system activation in schizophrenia. Inflammatory symptoms have been observed in post-mortem studies of patients with schizophrenia [54], and as such, inflammation might be a possible pathogenic mechanism in this disorder. Cyclooxygenase 2 (COX2), a rate-limiting enzyme in prostaglandin biosynthesis, mediates inflammatory responses and is induced by cytokines such as interleukins IL2, IL6, and IL10 [55]. By activating its canonical genomic pathway, VD_3_ down-regulates the expression of the *COX2/PTGS2* gene, which can inhibit pro-inflammatory cytokines secretion [55] (Figure 3).

## 4. VD_3_ and Autism Spectrum Disease

Autism Spectrum Disease (ASD) is one of the neurodevelopmental disorders that typically occurs in early childhood and is characterised by deficits in social interactions and communications, stereotypic and repetitive behaviour, limited interests, and impairments in sensory processing [56]. Although ASD aetiology is mostly unknown, a number of genetic and environmental factors, including VD_3_ deficiency, are considered to be associated with ASD [57]. Lower levels of VD_3_ have been reported in the brains of children and adolescents with ASD [58], and VD_3_ deficiency may aggravate the ASD symptoms in children [59]. Furthermore, studies investigating the impact of maternal VD_3_ deficiency on brain development found that children at the age of five showed more ASD-related symptoms, greater behavioural issues, and less social skills if there was prenatal calcidiol deficiency [60]. All these studies suggest a potential risk of low levels of VD_3_ in ASD development. A clinical intervention study was conducted to elucidate the effect of VD_3_ supplementation on ASD symptoms. In this study, mothers of their first child with ASD received VD_3_ supplementation during their second pregnancy. The result showed that the ASD prevalence of these children was reduced to 5% [61], which is much lower than the reported 18.7% from the older study [62]. A recent clinical study found thata daily VD_3_ dose of 300 IU/kg significantly improved the core symptoms of ASD children as mainly reflected in the Childhood Autism Rating Scale (CARS) scores, stereotypes, and greater eye contact and attention duration [63]. However, the manuscript describing this study was later retracted. Taken together, the currently available data cannot support the protective and therapeutic role of VD_3_ for ASD due to the lack of consensus methods, low number of interventional studies, and contradictory results.

## 5. VD_3_ and Attention Deficit Hyperactivity Disorder

Attention Deficit Hyperactivity Disorder (ADHD) is a neurodevelopmental disorder usually characterised by poor attention, hyperactivity, and impulsivity [64]. Researchers have postulated that genetic factors may contribute to up to 80% of cases [65], and neurological anomalies may also be responsible for the aetiology of ADHD [66]. Luo et al. reported that the deficit in spatial learning is strongly related to the neuropsychological characteristics of patients with ADHD [67]. Several studies on VD_3_ and neurodevelopmental diseases suggest that VD_3_ deficiency may be a risk factor for ADHD in children [68]. For instance, VD_3_-deficient mice show poor performance in hippocampal-dependent spatial learning. Apart from spatial learning, studies investigated whether experimental tasks used for studying attentional systems can be modulated by VD_3_ [69]. To this end, rat models with prenatal and postnatal VD_3_ deprivation were assessed by pre-pulse inhibition (PPI) of acoustic startle reflex in their adulthood. The results showed, despite having normal acoustic startle reflex, an impairment using PPI in these animals [69]. In the same study, it was also observed that rats with prenatal VD_3_ deficiency are hyperactive during the hole board test and the elevated plus maze test, and that VD_3_ depletion during pregnancy may display ADHD-like cognitive deficits [69]. However, there are also studies that have drawn contradictory conclusions. Orsini et al. [70] reported no differences in serum VD levels between controls and mice with a gene abnormality thought to be associated with ADHD. Therefore, it is still unclear if VD_3_ deficiency is associated with cognitive aberrations in ADHD. In one of the clinical studies, the VD_3_ status was tested in children and adolescents diagnosed with ADHD [71]. The findings from this study showed that study subjects had lower VD_3_ levels compared to matched controls [71]. Another study conducted by Tolppanen et al. [72] inspected the connection between VD_3_ status and the presence of childhood behavioural problems that are frequently seen in children with ADHD. However, no association has been seen between the VD_3_ status and the indicators of ADHD. Furthermore, impulsivity post-intervention was reduced in children with ADHD supplemented with 1000 IU per day of VD_3_ compared with placebo and matched control after three months. Nevertheless, there is no difference observed in attention and response inhibition [73]. All these data show that VD_3_ supplementation can reduce some, but not all, symptoms of ADHD, and may therefore be suggested as an addition to the currently used therapies [64].

## 6. VD_3_ and Alzheimer’s Disease and Dementia

Alzheimer’s Disease (AD) is one of the most widespread neurodegenerative diseases and the main cause of dementia [74]. AD is pathologically defined by the accumulation and aggregation of the amyloid-β (Aβ) peptides in extracellular space and the neurofibrillary tangles (NFTs) in intracellular compartments [75]. The phenotypic progression of the disease state is characterised by the functional decline of memory and cognition processes [75]. Many epidemiological reports showed the association between low VD_3_ concentration with an increased risk of AD and dementia when compared to matched controls [76,77,78,79]. Furthermore, recent studies indicate that serum hypovitaminosis of VD_3_ is linked with reduced hippocampal region volume in elderly AD patients connected to cognitive decline [80]. Apart from research regarding the direct serum VD_3_ concentrations, there are data showing a link between higher levels of serum calcidiol or higher baseline dietary VD_3_ intake with a lower risk of dementia and AD [81,82].

In addition to epidemiological studies based on VD_3_ intake, genetic research has also shown a connection between polymorphisms in the *VDR* or *low-density lipoprotein receptor-related protein 1* (*LRP1*) genes and risks of cognitive decline or AD (Figure 3). The study conducted by Gezen-Ak et al. [83] showed that single nucleotide polymorphism (SNP) at the *VDR Apa*I site increased the risk of AD by 2.3 times. Additionally, the *VDR* TaubF haplotype, which includes all together five SNPs, is significantly more prevalent in AD cases compared to healthy controls [84]. In addition, a recent study showed an association between *VDR Taq*I polymorphisms and AD susceptibility in both risk and protective factors [85]. These studies lend weight to the hypothesis that VD_3_-related pathways may be linked to the risk of developing AD.

As previously mentioned, the generation of Aβ depends on the sequential cleavage of APP protein by β-secretase (BACE), γ-secretase, and the subsequent degradation of Aβ via endopeptidases, such as neprilysin (NEP) and the insulin-degrading enzyme (IDE) [86]. Studies revealed that VD_3_ deficiency increases the high Aβ load by reducing the α-secretase and IDE levels while at the same time increasing γ-secretase in five Familial Alzheimer Disease (5xFAD) mice models [87]. Moreover, VD_3_ supplementation results in diminished Aβ load by decreased Aβ generation and increased Aβ degradation by reducing the expression level of BACE1 and γ-secretase [87,88]. Recent findings have highlighted the crucial role of LRP1 in VD_3_-mediated Aβ clearance. Interestingly, an in vitro study showed that calcitriol, an active form of vitamin D, can decrease the accumulation and uptake Aβ by up-regulating the *LRP-1* gene [89] (Figure 3).

AD is also associated with chronic neuroinflammation involving microglial and astrocytic changes. VD_3_ affects the microglial immune activation by increasing *IL10* expression [32], while VDR activation regulates microglia polarisation and oxidative stress [90]. These findings suggest the crucial role of VD_3_ in amyloid pathology. However, conflicting results have also been reported. Interventional research found that VD_3_ supplementation exerts no beneficial effect on AD development either alone or in combination with other nutrients [91,92]. Another study showed that APP/PS1 mice given a sufficient VD_3_ (600 IU/Kg) diet have shown lower serum calcitriol. Also, the supplementation of VD_3_ (8044 IU/kg) for 3 months worsens AD and increases Aβ deposition [93]. Consistently, VD_3_ supplementation for more than 146 days increases the risk of dementia by 1.8 in elder adults, and the risk of mortality in patients with dementia increased by 2.17 times compared to control groups [93]. However, additional research is required to determine whether VD_3_ exerts beneficial or adverse effects on AD.

## 7. VD_3_ and Parkinson’s Disease

Parkinson’s Disease (PD) is the second most common neurodegenerative disease after AD. It is characterised by symptoms including tremors, rigidity, and bradykinesia and is accompanied by the loss of dopaminergic neurons and Lewy pathology [94]. The role of VD_3_ in PD has been widely investigated. According to a recent systematic review and meta-analysis, PD patients had significantly lower serum calcidiol levels compared to healthy controls [95]. Out of eight observational studies examined in the systematic review, only one study showed no association between low serum calcidiol and PD [95]. Moreover, VD_3_ administration reduces the DA toxicity of 6-hydroxydopamine [96] (Figure 3), and the lack of VDR induces a significantly impaired motor function in mice [97]. Furthermore, the link between PD and *VDR* gene polymorphisms has also been explored. One study found multiple SNPs were associated with PD (Figure 3), suggesting the *VDR* as a potential susceptibility gene for PD [98].

Some studies have investigated the use of VD_3_ as a treatment to reduce some of the PD symptoms. Suzuki et al. [99] evaluated the relationship between 1200 IU per day of VD_3_ and disease progression for two years follow-up in a randomised, double-blind, placebo-controlled trial. As expected, those who received VD_3_ supplements were more likely to have higher serum calcidiol and were significantly less likely to have any deterioration of PD as measured with the Hoehn and Yahr scale (H&Y) and Unified Parkinson’s Disease Rating Scale (UPDRS) [99]. Patients with PD receiving placebo had a worse neurological outcome. Luthra et al. [100] conducted a cohort study in early PD patients followed for three years and divided into three groups: supplementation with multivitamin (MVI), VD_3_ administration, and co-treatment with VD_3_ and MVI. However, the authors did not observe differences in disease progression within the three groups [100].

## 8. VD_3_, and Epilepsy and Seizures

Epilepsy is a type of neurological disorder characterised by symptoms of periodic, unpredictable, and recurrent seizures [101]. Although the cause of epilepsy is still unknown in most cases, seizures can be caused by any damage associated with metabolic disorders, infectious diseases, genetic mutation, or immune disorders that disrupt brain function [102]. The potential role of VD_3_ in the pathophysiology and treatment of epilepsy has received less attention in recent years. The VD_3_ deficiency is prevalent in patients with epilepsy with 70–80% of them having low levels of serum calcidiol [103]. An early study showed that a direct administration of calcitriol to the hippocampus or intravenous injection of large doses increased the threshold for seizure activity in rats [104]. The pre-administration of calcitriol in mice was also shown to have anti-convulsant effects, as it again increased the electroconvulsive threshold and potentiated the effects of the well-known anti-convulsant drugs sodium valproate and phenytoin [105]. Furthermore, a small controlled pilot study showed a 40% reduction in seizures following the treatment with VD_3_ compared to placebo [106]. The plausible mechanism for VD_3_’s protective role may be the regulation of Ca^2+^ signalling in neurons through the modulation of LVSCC (Figure 3). VD_3_ alters the conductance of LVSCC and chloride channels, therefore affecting the neuronal excitability and susceptibility to seizures at the threshold level [107]. VD_3_ lowers the expression of certain pro-convulsant cytokines, such as *IL1B* and *TNF*, and increases the expression of anti-convulsant growth factors *NTF3* [107] (Figure 3).

## 9. VD_3_ and Amyotrophic Lateral Sclerosis

Amyotrophic Lateral Sclerosis (ALS) is a fatal neurodegenerative disease that is characterised by the progressive degeneration of motor neurons in the CNS, which leads to muscle weakness, paralysis, and death [108]. Regarding the role of VD_3_ in ALS, the studies reported very contradictory results, going from protective [109] to detrimental [110]. Karam et al. [111] observed VD_3_ deficiency in patients with ALS and decreased ALS-FRS-R scores in 20 patients after supplementation with VD_3_ for 9 months at 2000 IU per day [111]. Recent investigations on the ALS G93A transgenic mouse model found that VD_3_ administration enhanced functional outcomes when compared to control mice [112,113]. However, one recent finding highlighted that 6-month supplementation of VD_3_ in ALS patients had no significant effects on motor dysfunction [114]. Moreover, there is also evidence showing the negative effect of higher VD_3_ levels on prognosis in ALS [110].

Evidence suggests that VD plays an important role in maintaining muscle strength and the regeneration of muscle damage. In physically active healthy older adults, VD_3_ was shown to prevent muscle weariness by regulating the production of creatine kinase, lactic acid dehydrogenase, troponin I, and hydroxyproline [115]. Additionally, VD_3_ may influence the proliferation and differentiation of muscle precursors to impact muscle function [116].

## 10. VD_3_, and Headaches and Regulation of Pain

Headache is among the major neurological disorders in adults and children, impacting their quality of life. Headaches can be primary or secondary; the primary headaches account for almost 98% of all headaches and are not potentially life-threatening [117]. Some studies have shown a strong relationship between primary headaches, especially migraine, and VD_3_ deficiency in children [118,119] as well as in adults [120]. This association is also supported by the increased prevalence of headaches in autumn and winter at high latitudes, where VD_3_ plasma levels are generally lower in people [121]. There is evidence indicating that VD_3_ deficiency can cause pain. In a meta-analysis of 81 observational studies, low VD_3_ concentration was connected with arthritis, muscle pain, and chronic widespread pain [122]. VD_3_ supplementation may play a role in the treatment of headaches with a basic focus on the prevention of headache attacks. A recent randomised, placebo-controlled parallel trial of VD_3_ supplementation in patients with migraine showed a significant decrease in migraine frequency from baseline to week 24 compared with placebo. The number of headache days was reduced by the end of the trial in those taking VD_3_; nonetheless, there was no significant change in the migraine severity, pressure pain thresholds, or temporal summation [123]. In addition, Cayir et al. [124] found that the inclusion of VD_3_ in current anti-migraine treatment with amitriptyline reduced the number of migraine attacks in paediatric migraine patients compared to the group receiving amitriptyline treatment alone [124]. However, a Cochrane review published in 2015 showed VD supplementation may not be better than placebo to control chronic pain in adults [125].

There are possible ways by which VD_3_ may influence primary headaches. The anti-inflammatory role of VD_3_ plays a key role in migraine, since inflammatory substances can activate the trigeminal nerve, which is the main structure involved in migraine headaches [126] (Figure 3). One of the most important mechanisms by which VD_3_ deficiency could contribute to headache is through the possible sensitisation of the second and third neurons, which are connected with the stimulation of sensory receptors of the periosteal covering and central sensitisation [127]. Another mechanism underlying the protective effect of VD_3_ in both migraine initiation and maintenance involves the direct relationship between VD_3_ and magnesium (Mg^2+^) serum concentrations. Indeed, VD_3_ promotes the intestinal absorption of Mg^2+^, which has been demonstrated to be effective against chronic pain conditions and migraine by contrasting vascular and neurogenic mechanisms [128,129]. VD_3_ also influences the release of DA and serotonin, which is known to be connected with the pathogenesis of migraine [130]. In particular, VD_3_ can affect the synthesis of serotonin via the key enzyme TH. Hence, in addition to its role in migraine pathogenesis, VD_3_ deficiency may also cause depression, which often coexists with all types of headaches [125]. Moreover, VD_3_ can influence the modulation of pain by regulating the synthesis of neurotrophic factors, including NGF [131] (Figure 3).

## 11. VD_3_ and Sleep Disorders

Sleep disorder, also known as sleep–wake disorder, refers to a group of conditions that affect the quality, timing, and amount of a person’s sleep, including insomnia, sleep apnoea, restless leg syndrome, and narcolepsy [132]. In recent years, there has been increasing interest in the potential role of VD_3_ in sleep regulation. Studies have shown that VD_3_ deficiency is associated with an increased risk of sleep disorders. For instance, a cross-sectional study within a large cohort of elderly men found that VD_3_ deficiency was related to poorer sleep quality [133]. On the other hand, McCarty et al. reported that individuals with higher levels of VD_3_ had better sleep quality than those with lower VD_3_ levels [134]. Additionally, several epidemiology studies have observed that VD_3_ supplementation may improve sleep quality. A double-blind clinical trial reported improvements in sleep duration and quality in people with sleep disorder after taking 50,000 IU/fortnight VD_3_ for 8 weeks [135]. Another case study showed that VD_3_ intake (50,000 IU/week) improved sleep latency and duration in veterans with sleep problems [136]. However, a cross-sectional study reported no significant difference in the Pittsburgh Sleep Quality Index (PSQI) total score between pregnant women in their last trimester with inadequate VD_3_ levels and those with adequate VD_3_ levels [137]. Moreover, several randomised controlled trials reported no improvement in sleep disorders after VD_3_ supplementation for 12 weeks [138], 18 months [139], or even as long as 5.3 years [140] compared to the control group.

The mechanisms through which VD_3_ may regulate sleep are not fully understood. However, several studies have proposed potential mechanisms that may explain the relationship between VD_3_ and sleep. One plausible mechanism is that VD_3_ plays a role in down-regulating the transcription of the *v-rel avian reticuloendotheliosis viral oncogene homolog B proto-oncogene NFκB RELB* gene through VDR signalling [141]. The *RELB* gene is one of the members of the NFκB family that plays a role not only in the production of sleep-regulating substances, such as IL-1 and TNF, but also in the selective activation of inflammatory pathways known to occur in the setting of intermittent hypoxia, such as in obstructive sleep apnoea [142]. Another possible mechanism is related to the role of VD_3_ in the regulation of the serotonergic pathway and in melatonin production [5,143]. VD_3_ can be involved in regulating the expression of *tryptophan hydroxylase* genes (*TPH1* and *TPH2*), which are responsible for the conversion of tryptophan into serotonin in the brain compared to other tissues, by binding to the VDRE located in the promoters of these genes [143]. VD_3_ has been shown to increase *TPH2* expression in the brain [143]. Moreover, VD_3_ plays a significant role in sleep by influencing *THP1* expression in the pineal gland. The pineal gland transforms serotonin into melatonin during the evening and night through the expression of *THP1* [144].

## 12. VD_3_, and Depression and Bipolar Disorder

Depression and Bipolar disorder are complex psychiatric conditions influenced by genetic, environmental, and biochemical factors [145,146]. Emerging evidence suggests that vitamin D deficiency may play a significant role in the pathogenesis of these disorders [147,148,149].

As discussed previously, vitamin D is involved in the regulation of several crucial processes in the brain, including neurotransmitter synthesis, neuroinflammation, neuroplasticity, and neuroprotection. Vitamin D also contributes to the synthesis of neurotransmitters such as serotonin, dopamine, and norepinephrine, which are essential for mood regulation [130,143,150]. Deficiencies in these neurotransmitters are commonly observed in both depression and bipolar disorder [151,152,153]. Furthermore, vitamin D has anti-inflammatory properties that can help reduce neuroinflammation, which is a condition often linked to the pathogenesis of these mental health disorders. By inhibiting the production of pro-inflammatory cytokines, vitamin D can mitigate chronic inflammation in the brain [32,33]. Additionally, vitamin D promotes the expression of neurotrophic factors such as BDNF, which are crucial for neuroplasticity and neuronal survival [20]. Impaired neuroplasticity is a hallmark of depressive and bipolar disorders, and enhancing BDNF expression may offer therapeutic benefits [154,155]. Vitamin D also modulates the hypothalamic–pituitary–adrenal (HPA) axis, which regulates the body’s response to stress [156,157]. Dysregulation of the HPA axis is often observed in individuals with depression and bipolar disorder, suggesting that vitamin D may help restore balance to this system [156,158,159].

Many studies have investigated the role of vitamin D in bipolar disorder with mixed results. Humble et al. [160] found deficient vitamin D levels (48 nmol/L) in 22 bipolar disorder patients within a psychiatric cohort, but these were not statistically significant compared to other groups. Similarly, Menkes et al. [161] noted that all psychiatric patients, including 19 with bipolar disorder, had vitamin D levels below 50 nmol/L, except for the schizophrenia group, which had markedly lower levels. Lapid et al. [162] also found no significant differences in vitamin D levels among 141 psychiatric inpatients, including 11 with bipolar disorder. Boerman et al. [163] found no significant vitamin D level differences between large cohorts of bipolar disorder and schizophrenia patients after adjusting for confounding variables. Grønli et al. [164] observed significantly lower vitamin D levels in 95 psychiatric patients (10 with bipolar disorder) compared to healthy controls, but no significant differences among different psychiatric diagnoses, and Belzeaux et al. [165] reported more severe vitamin D deficiency in mood disorder patients compared to those with schizophrenia with bipolar disorder patients showing low levels similar to those with major depressive disorder. Altunsoy et al. [166] found lower vitamin D levels in bipolar disorder patients during acute manic episodes compared to healthy controls with a negative correlation between vitamin D levels and manic symptom severity. Interestingly, Petrov et al. [167] found higher vitamin D-binding protein levels in bipolar disorder adolescents compared to those with non-major mood disorders, suggesting it as a potential biomarker for bipolar disorder.

Clinical interventions have explored the efficacy of vitamin D supplementation in managing depression and bipolar disorder with mixed results. Some studies have shown significant improvements in mood and symptom reduction following vitamin D supplementation, especially in individuals with clinically significant vitamin D deficiency [168]. However, contrasting findings indicate that other studies have reported no observed improvement [169]. Nevertheless, the potential benefits of vitamin D in clinical management include its role in enhancing the effects of conventional treatments and providing an adjunctive therapy for individuals who do not fully respond to standard medications.

In the case of depression, supplementation with vitamin D has been associated with mood improvements, which is possibly due to its effects on serotonin synthesis [170] and neuroinflammation reduction [171]. However, the optimal dosage of vitamin D for managing depression is still under investigation with some studies suggesting doses ranging from 800 to 2000 IU/day [172].

For bipolar disorder, the evidence is less extensive, but the mechanisms through which vitamin D influences neurotransmitter regulation, neuroinflammation, and neuroplasticity may also be relevant. Despite Marsh et al. [147] finding no significant improvement in mood among 33 vitamin D-deficient bipolar disorder patients despite increased vitamin D levels with supplementation, Sikoglu et al. [173] observed that 8-week vitamin D_3_ supplementation in young bipolar disorder patients reduced manic and depressive symptoms and increased GABA concentrations in the brain. While more robust clinical trials specifically targeting bipolar disorder are needed, preliminary studies suggest that vitamin D supplementation could potentially stabilise mood swings and alleviate the severity of depressive episodes.

## 13. Conclusions and Future Prospects

VD_3_ is essential for many physiological processes, including the regulation of the immune system and the development and function of the adult brain (Figure 4). As a result, VD_3_ has undergone extensive investigation in a wide array of pathological conditions either as a contributing factor or as a blood serum biomarker associated with disease severity. For example, epidemiological studies have linked low neonatal VD_3_ levels to developmental brain diseases such as ASD, ADHD, and schizophrenia. Currently, there is established recognition regarding the correlation between insufficient VD_3_ levels and various neurodegenerative diseases like AD, PD, ALS, and dementia. This awareness has underscored the significance of VD_3_ supplementation and the need to maintain adequate plasma levels as a potential means of preventing or alleviating certain neurological diseases. However, it is worth noting that a limitation of these studies is that the majority of the research primarily focuses on the improvement of symptoms without addressing the potential cognitive enhancements that may be associated with VD_3_.

The aforementioned points shed light on an important limitation and raise pertinent questions that warrant further exploration and investigation:**Cognitive Augmentation**: It is crucial to explore whether VD_3_ has any direct impact on cognitive function, including memory, attention, and executive functioning. Further research is needed to elucidate the potential cognitive benefits of VD_3_ and its underlying mechanisms.**Optimal Dosage and Duration**: Determining the optimal dosage and duration of VD_3_ supplementation is essential for maximising its potential preventive or ameliorative effects in various neurological diseases. Studies should aim to investigate the dose–response relationship and the duration required to achieve significant clinical outcomes. Moreover, exploring potential variations in optimal dosage and duration across different age groups and populations is vital to tailor interventions effectively.**Long-Term Effects**: Understanding the sustained benefits, as well as any potential risks or adverse effects associated with prolonged VD_3_ use, is crucial for establishing guidelines and recommendations. Longitudinal studies are warranted to assess the impact of VD_3_ supplementation on disease progression, overall health, and cognitive aging.**Clinical Trials and Intervention Studies**: Well-designed clinical trials and intervention studies are needed to establish causality and determine the efficacy of VD_3_ supplementation as a preventive or therapeutic strategy. Randomised controlled trials, employing appropriate control groups and standardised outcome measures, will provide robust evidence for guiding clinical practice.**Reverse Causality**: The potential reverse causality, where a particular disease may decrease VD_3_ levels, should be studied to understand the relationship between VD_3_ levels and neurological diseases. Factors such as nutrition, exercise, and sun exposure should also be considered in linking low VD_3_ levels with neurological diseases.**Mechanistic Insights**: Gaining a deeper understanding of the underlying molecular and cellular mechanisms through which VD_3_ influences cognitive function and disease progression is essential. Investigating VD_3_’s interactions with specific receptors, signalling pathways, and gene expression patterns can provide valuable insights into its therapeutic potential and aid in the development of targeted interventions.**Gene Regulatory Potential**: Further studies are needed to explore the gene regulatory potential of VD_3_ in both developing and adult brains. Understanding the epigenetic control of VD_3_-regulated target genes and its implications for brain development and function can provide valuable insights.

Addressing these limitations and answering the associated questions will contribute to a comprehensive understanding of the potential benefits and optimal use of VD_3_ in various pathological conditions, particularly in terms of cognitive enhancement and disease prevention.

## Figures and Tables

**Figure 1 ijms-25-07809-f001:**
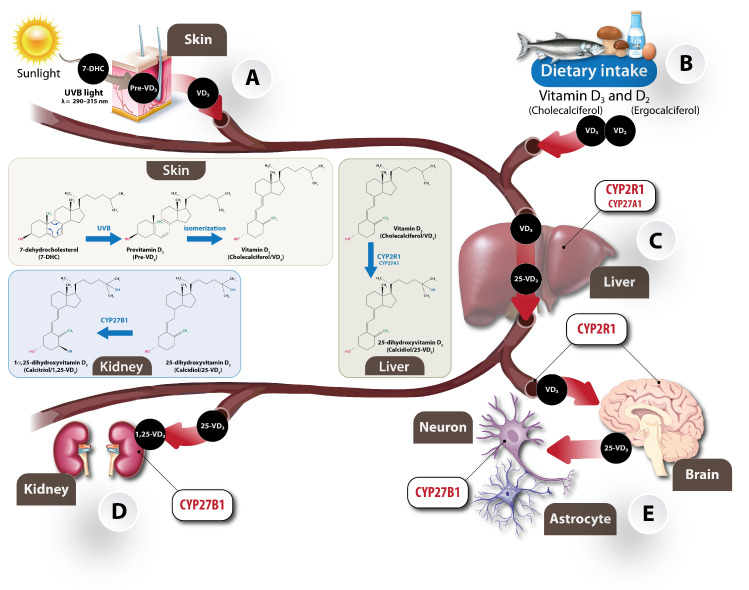
Sources of vitamin D_3_ and pathways for calcitriol biosynthesis in the body. (**A**) The classical pathway involves the synthesis of VD_3_ in the skin under the UVB radiation from the sun. This pathway starts with 7-dehydrocholesterol (7-HDC), a derivative of cholesterol, reacting to UV radiation to form cholecalciferol (VD_3_) through an intermediate previtamin D_3_ (Pre-VD_3_). (**B**) The alternative pathway involves dietary intake of VD_3_ or ergocalciferol (VD_2_)-rich food such as fatty fish, eggs, or fortified milk products for VD_3_ or plants and fungi for VD_2_. Under normal circumstances, higher quantities are synthesised via classical pathways. VD_3_ from both sources enters the circulation, binds to vitamin D binding protein (VDBP), and is transported to metabolic tissues. (**C**) The traditional synthetic pathway of 1α,25-dihydroxyvitamin D_3_ (calcitriol/1,25-VD_3_), the most active metabolite of vitamin D, starts in the liver when VD_3_ is hydroxylated at C25 by CYP2R1 (and in minority also CYP27A1) to yield 25-dihydroxyvitamin D_3_ (calcidiol/25-VD_3_), which is the major circulating storage form of vitamin D. (**D**) This 25-VD_3_ is transported by VDBP to the kidney, where it is hydroxylated at position C1 by CYP27B1 to form 1α,25-VD_3_. The 1α,25-VD_3_ binds to its molecular target vitamin D receptor (VDR), which regulates the transcription of various target genes. (**E**) Vitamin D can also be transported directly to the brain as 25-VD_3_ or 1α,25-VD_3_, where both can cross the blood–brain barrier. Additionally, 25-VD_3_ can be converted to 1α,25-VD_3_ in the brain, since the enzymes involved in its synthesis are expressed in pericytes, glial cells, and neurons in addition to the liver and kidney. This suggest the possible of local synthesis of vitamin D metabolites in the brain and active signalling.

**Figure 2 ijms-25-07809-f002:**
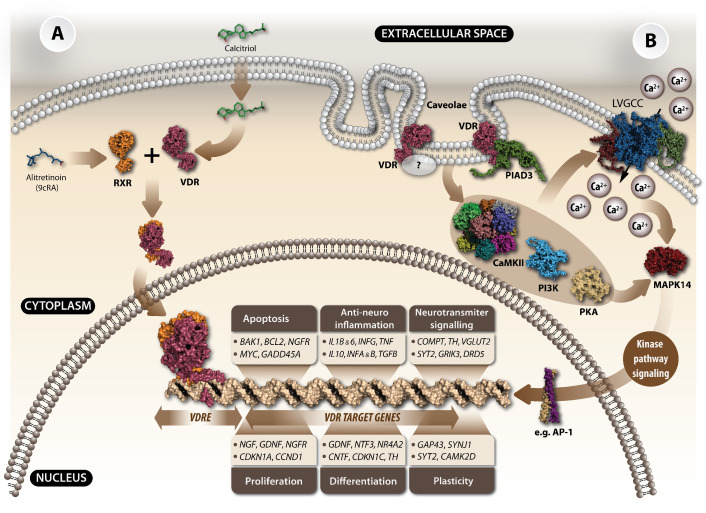
Schematic model illustrating the genomic and non-genomic effects of VD_3_ in the CNS. (**A**) Genomic action of calcitriol: This model focuses on VD_3_ target genes and signalling pathways that have been identified within the brain or neural cells (neurons, astrocytes, oligodendrocytes, or microglia). Genomic actions primarily occur within the nucleus. In this simplified model, calcitriol binds to the VDR/RXR complex, leading to the release of co-repressors and the recruitment of co-activators at vitamin D response elements (VDREs) located in regulatory regions, which promotes the expression of specific calcitriol target genes. The listed genes are those whose expression is influenced by VD_3_ within the brain and for which functional VDREs have been identified on relevant regulatory regions. (**B**) Non-genomic action of VD_3_: The non-genomic actions of _3_ are potentially mediated through the classical VDR, protein disulphide isomerase A3 (PDIA3), or both of these proteins. Upon calcitriol binding, the rapid activation of protein kinases such as CaMII, PKA, and PI3K occurs, which in turn facilitates the influx of Ca^2+^ ions via L-type voltage-gated calcium channels (L-VGCCs). Intracellular Ca^2+^ then triggers the activation of p38MAPK, further modulating downstream signalling pathways. Both the genomic and non-genomic actions of VD_3_ are likely to have an impact on brain development, function, and maintenance. These mechanisms play a role in shaping the intricate processes within the CNS.

**Figure 3 ijms-25-07809-f003:**
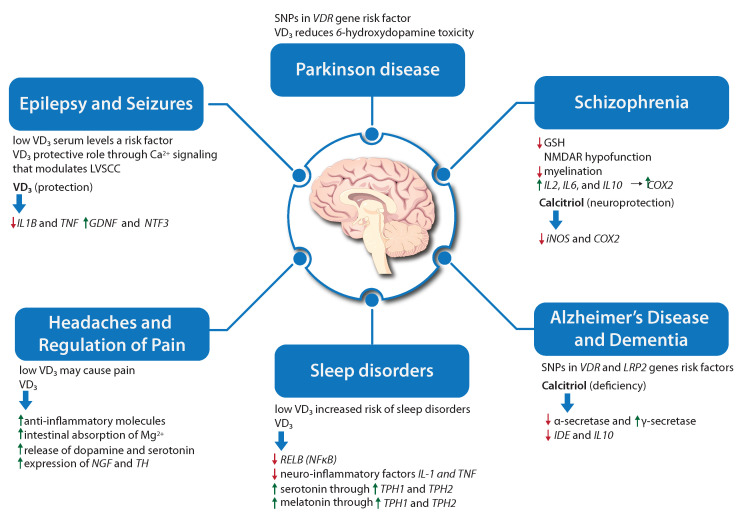
The connection between VD_3_ and various neurological disorders is summarised in the figure. It highlights diseases in which molecular connections have been suggested between VD_3_ signalling and gene regulatory or metabolic networks, such as the regulation of enzymes that facilitate the production of serotonin or melatonin. Additionally, a general neuroinflammation process has been proposed as a link to many neurological diseases, where VD_3_ supplementation inhibits the expression of inflammatory cytokines and/or activates the expression of anti-inflammatory molecules. Throughout the figure, the following abbreviations have been used: glutathione (GSH), *interleukin-1β* (*IL1B*), *tumour necrosis factor* (*TNF*), *L-type voltage-sensitive Ca^2+^* channel, *glial cell-derived neurotrophic factor* (*GDNF*), *neurotrophin 3* (*NTF3*), *nerve growth factor* (*NGF*), *tyrosine hydroxylase* (*TH*), *tryptophan hydroxylase* (*TPH1* and *TPH2*), *N-methyl-D-aspartate receptor* (*NMDAR*), *inducible nitric oxide synthase* (*iNOS*), *prostaglandin-endoperoxide synthase 2 or cyclooxygenase-2* (*COX2/PTGS2*), *insulin-degrading enzyme* (*IDE*), *low-density lipoprotein receptor-related protein 1* (*LRP1*), *vitamin D receptor* (*VDR*).

**Figure 4 ijms-25-07809-f004:**
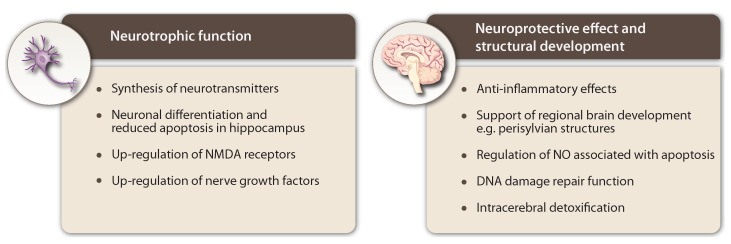
Overview of the effects of VD_3_ on individual neurons and the whole brain.

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
