# Peer review of "Vitamin D in Central Nervous System: Implications for Neurological Disorders"

_ijms, 2024, doi:10.3390/ijms25147809_

Round 1
Reviewer 1 Report
Comments and Suggestions for Authors
Dear Authors the review article is quite interesting, and catchy for the readers, but I have few questions, positive and negative please address it.
1. The review delves into the importance of vitamin D in the brain and neurological diseases, including topics such as its functions in developing neurons, regulating the dopaminergic system, and producing neural growth factors.
2. It also assesses the possible advantages of vitamin D supplementation and the effects of vitamin D insufficiency on neurodegenerative disorders.
3. To fully grasp the therapeutic benefits of vitamin D for neurological illnesses, the article stresses the necessity for additional well-designed trials.
4. It adds to our knowledge of the possible advantages and best way to use VD3 in different pathological situations.
5. The vast majority of studies on VD3 have concentrated on alleviating symptoms rather than investigating any possible cognitive benefits.
6. There has been little research on the impact of VD3 on cognitive enhancement, which is a major area that needs further attention.
7. The article may benefit from a more even-handed presentation of the pros and cons of VD3 supplementation for neurological disorders.
Author Response
Please kindly find the reply in the attached PDF file.
Thank you.
Sincerely,
Ferdinand Molár

Reviewer 2 Report
Comments and Suggestions for Authors
The manuscript “Vitamin D in central nervous system: implications for neurological disorders .” by Bayan Sailike et al. aimed to investigate the relationships between vitamin D and central nervous system and investigate the consequences of vitamin D deficiency in neurological disorders, particularly neurodegenerative diseases. Comments:
1). Since there are several studies in the literature on the relationships between Vitamin D and Depression and Bipolar Disorder,should add a separate paragraph that examines the role of Vitamin D in the pathogenesis and clinical management of depression and bipolar disorder.
2). Page 1; Line 21:the Authors should point out that there are also other types of cells capable of completing the synthesis of Vitamin D and its receptor, such as Myocytes
3).The Authors should clarify whether the studies were conducted with cholecalciferol alone or also with calcifediol and whether there are differences between these two different types of Vitamin D in interacting with the cells of the central nervous systemc
Comments on the Quality of English LanguageThe quality of English Language is adequate
Author Response
Please kindly find the reply in the attached PDF file.
Thank you.
Sincerely,
Ferdinand Molnár
